# Abrasive Wear and Physical Properties of In-Situ Nano-TiC$_x$ Reinforced Cu–Cr–Zr Composites

**Dongdong Zhang** [1,2,*], **Pengyong Lu** [1], **Xiya He** [1] **and Yali Gao** [1,*]

[1] School of Mechanical Engineering, Northeast Electric Power University, No. 169 Changchun Road, Jilin 132012, China; lu971001@163.com (P.L.); 2202000724@neepu.edu.cn (X.H.)

[2] Gongqing Institute of Science and Technology, No. 1 Gongqing Road, Gongqing 332020, China

[*] Correspondence: zhangdongdong@neepu.edu.cn (D.Z.); dehuigyl@126.com (Y.G.)

**Abstract:** Cu–Cr–Zr alloys reinforced in situ with TiC$_x$ nanoparticles were prepared via combustion synthesis and electromagnetic stirring casting. The microstructure of TiC$_x$/Cu-Cr-Zr composites with various contents was analyzed. The microhardness and Brinell hardness of the composites were determined; the average volumetric abrasive wear rate and worn surface of the composites were investigated; and the electrical, thermal conductivity and thermal expansion coefficients of the materials were discussed. The results indicated that the addition of TiC$_x$ particles transformed the Cu–Cr–Zr matrix alloy microstructure from a dendritic to an equiaxed crystal, and the grain size was significantly refined as the amount added was increased. The composites with high TiCx content possessed higher hardness and abrasive wear resistance. The addition of TiC$_x$ particles reduced the electrical and thermal conductivity and thermal expansion coefficients of the materials.

**Keywords:** TiC$_x$/Cu; in situ nano-TiC$_x$; abrasive wear resistance; physical properties

## 1. Introduction

With the rapid development of the automotive, electronics, and machinery manufacturing industries [1–3], higher performance requirements have been put forward for Cu; that is, to ensure that Cu has good electrical and thermal conductivity on the basis of high strength, high wear resistance, especially with good high-temperature mechanical properties, and high temperature resistance to melt corrosion [4–6]. Adding alloying elements to a metal enables the physical and mechanical characteristics of the material to be improved, which expands the range of applications for metal materials [7,8]. For example, the electrode materials used in spot welding robots in passenger car manufacturing are Cu-based alloys, such as Cu–Cr–Zr, Cu–Cr and Cu–Zr, which have good electrical and thermal conductivities [9,10]. However, Cu alloys present many shortcomings in terms of strength and wear resistance, especially as electrode materials, due to the harsh working conditions and poor oxidation resistance, low strength and wear resistance, softening and deformation at temperatures above 500 °C, and melt corrosion problems, which result in large contact resistance, rise in conductive temperature, and serious loss of the electrode [11,12]. Existing spot welding electrodes have a very short service life of around only 500 cycles. Therefore, it is imperative to obtain a Cu alloy material with high strength, good wear resistance, and good thermal and electrical conductivity simultaneously [13,14].

It is known that the strength and wear resistance of composites can be effectively improved by incorporating nano-sized particles into the metal matrix [15]. Furthermore, ceramic particle morphology significantly impacts the mechanical and physical properties of the materials [16–18]. Cubic TiC$_x$ particles added in situ can enhance both electrical conductivity and the compression strength and yield strength of the materials. Nevertheless, the concentrated stresses and shear effects at the sharp corners of cubic TiC$_x$ particles substantially reduce the fracture strain of the composite materials. Compared with cubic TiC$_x$, the addition of spherical TiC$_x$ particles enhances the strength of the material and

avoids the reduction in fracture strain caused by the sharp corners of cubic $TiC_x$. Combustion synthesis is usually used to prepare spherical $TiC_x$ nanoparticle-reinforced Cu matrix composites with regular morphology and uniform size [19–21]. However, low-content $TiC_x$ particle-reinforced Cu matrix composites cannot be prepared by combustion synthesis owing to the limitations of combustion synthesis thermodynamic conditions [22,23]. Excessive $TiC_x$ content would substantially reduce electrical and thermal conductivity as well as the plasticity of the Cu matrix, making Cu worthless for its applications. Therefore, there is an urgent needed to develop a new method for manufacturing Cu matrix composites strengthened with a low content of $TiC_x$ particles. The application of casting methods makes it possible to add small amounts of reinforcing phases and alloying elements into the melt [24–26]. Therefore, low-content, spherical $TiC_x$ nanoparticle in situ-strengthened Cu–Cr–Zr-based composites can be fabricated organically via the combination of combustion synthesis method with stir casting.

In this paper, $TiC_x$/Cu master alloys with a reinforcing particle size of about 100 nm and spherical particle shape were prepared by combustion synthesis; the $TiC_x$/Cu master alloys were then added and dispersed into the Cu–Cr–Zr melt using electromagnetic stirring casting to fabricate low-content $TiC_x$/Cu–Cr–Zr composites. The experiments focused on the impact on microstructure, abrasive wear resistance, electrical and thermal conductivity, and thermal expansion coefficients of the composites with various contents of $TiC_x$ nanoparticles. The mechanism responsible for the effect of the in situ incorporation of spherical $TiC_x$ nanoparticles on the variation of microstructure, abrasive wear resistance, and the above-mentioned physical properties of the composites was revealed. A new method of preparing $TiC_x$/Cu–Cr–Zr composites is presented in this paper. Furthermore, the results will provide the experimental foundation and theoretical guidance for further investigation of both the microstructure and properties of $TiC_x$ ceramic particle-reinforced Cu-based alloys.

## 2. Materials and Methods

Cu powders with an average size of 45 μm, Ti powders with an average size of 25 μm, and CNTs with a length of 15–80 μm and a diameter of 10–20 nm were used to fabricate $TiC_x$/Cu master alloys by the combustion synthesis method. The $TiC_x$/Cu–Cr–Zr composites were manufactured via the method of combustion synthesis assisted by electromagnetic stirring casting; details of the process flow are shown in Figure 1. The experimental powder was configured with a CNT-Ti molar ratio of 1.0 and a Cu content of 70 vol.%. After applying the ball mill to mix the configured powder, the resulting mixed powder was cold-pressed at 100 MPa into a cylindrical compact measuring 30 mm in diameter and 30 mm in height. The resulting compact was placed in graphite mold in a homemade vacuum furnace, which was evacuated to below 100 Pa and heated. According to the vacuum gauge indication, heating was stopped after the combustion synthesis reaction, which resulted in the $TiC_x$/Cu master alloy. Manufacturers of Ti, Cu, CNTs, and Cu–Cr–Zr alloy are shown in Table 1. The chemical composition of the Cu–Cr–Zr alloy is illustrated in Table 2. The Cu–Cr–Zr alloy was heated to a molten state in the graphite crucible of an electromagnetic induction furnace for incorporation of the $TiC_x$/Cu master alloy into the molten state Cu–Cr–Zr. During electromagnetic stirring, mechanical stirring was used as an aid for dispersion of the $TiC_x$/Cu master alloy. After the master alloy was completely dispersed, it was held at this temperature for 3 min and then cast into the prepared mold. Successful $TiC_x$/Cu–Cr–Zr composites were fabricated by cooling to room temperature.

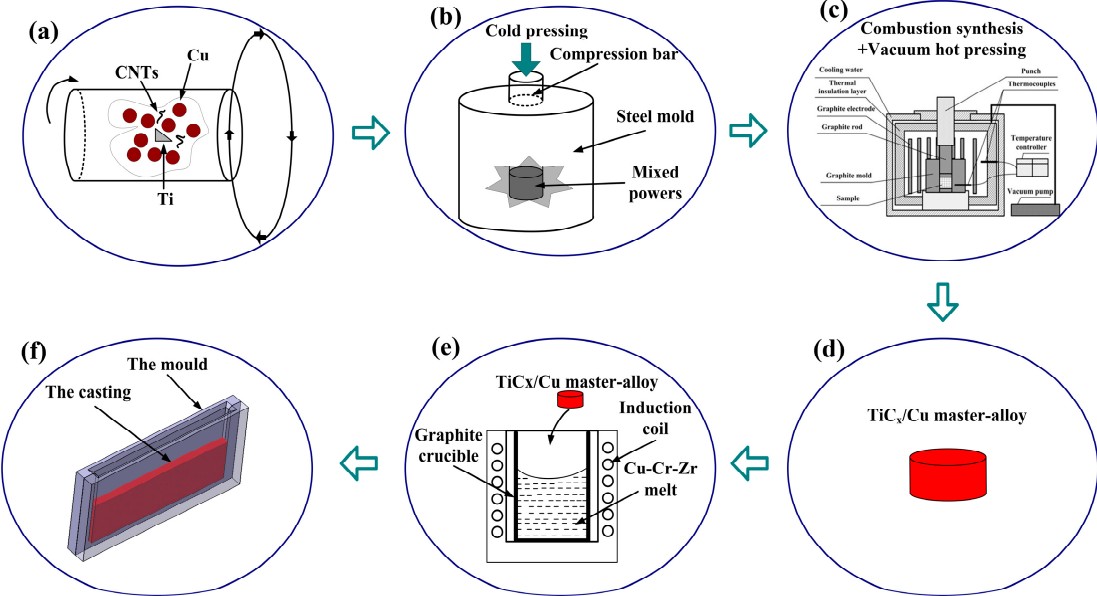

**Figure 1.** Flow chart for the preparation of TiC$_x$/Cu–Cr–Zr composites utilizing combustion synthesis-assisted stir casting. (**a**) ball milling (**b**) cold pressing (**c**) combustion synthesis + hot press (**d**) master-alloy (**e**) casting process (**f**) casting.

**Table 1.** Manufacturers of raw materials.

| Raw Material | Manufacturer |
|---|---|
| Ti | Beijing Research Institute of Nonferrous Metals |
| Cu | Beijing Research Institute of Nonferrous Metals |
| CNTs | Chengdu Organic Chemistry Co., Ltd., Chinese Academy of Sciences (Chengdu, China) |
| Cu–Cr–Zr | Shanghai Xinmeng Metal Materials Co., Ltd. (Shanghai, China) |

**Table 2.** Chemical compositions of Cu–Cr–Zr alloy (wt.%).

| Elements | Cu | Cr | Zr | Si | Fe | Mg | Al |
|---|---|---|---|---|---|---|---|
| Content | Bal. | 0.65 | 0.6 | 0.05 | 0.05 | 0.15 | 0.18 |

　　　　TiC$_x$/Cu master alloy phase composition was analyzed on a Rigaku D/Max 2500PC X-ray diffraction (XRD, Rigaku D/Max 2500PC, Tokyo, Japan) with Cu-K$\alpha$ radiation. Morphologies of microstructure and extracted TiC$_x$ were investigated using a field emission scanning electron microscope (FESEM, JSM 6700, Tokyo, Japan). Observation of the grain of TiC$_x$/Cu–Cr–Zr composites was performed using an Olympus optical microscope (XJZ-6, Tokyo, Japan). Determination of the microhardness for materials using a Vickers hardness tester (1600-5122VD, New Troy, MI, USA) with 50 gf load and 15 s duration time. Measurement of the Brinell hardness was performed using a Brinell hardness tester (HB-3000C, Kunshan, China) with 9807 N load and a duration time of 30 s. Both microhardness and Brinell hardness measurements were collected 5 times and averaged. Tests of abrasive wear were performed using a pin-disc machine with Al$_2$O$_3$ abrasive paper under a load of 5 N with a wear distance of 24.78 m at room temperature. The samples were 4 mm in length, 4 mm in width, and 12 mm in height. The worn surfaces of the composites were observed by a scanning electron microscope (SEM, Evo18, Carl Zeiss, Jena, Germany). A digital eddy current conductivity meter (Sigma 2008b, Kanagawa, Japan) was employed to determine the electrical conductivity; the results were measured 5 times and averaged. A laser thermal conductivity meter (LFA427, Selb, Germany) was employed to measure the thermal diffusivity at the temperature of 25 °C, 50 °C, 75 °C, and 100 °C. Coefficient of

thermal conductivity was defined as the product of thermal diffusivity, density, and specific heat. The coefficient of thermal expansion was tested using a dilatometer (NETZSCH DIL402C, Selb, Germany) in argon at the temperatures of 50 °C, 100 °C, 150 °C, 200 °C, 250 °C, 300 °C, 350 °C, and 400 °C.

## 3. Results and Discussion

The X-ray diffraction pattern of $TiC_x$/Cu master alloy prepared through combustion synthesis is illustrated in Figure 2a. The diffraction peaks belonging to Cu and $TiC_x$ were clearly visible, demonstrating that the $TiC_x$/Cu master alloy was successfully manufactured. The morphology of $TiC_x$ nanoparticles extracted from the $TiC_x$/Cu master alloy is depicted in Figure 2b. As indicated, the extracted $TiC_x$ particles from the $TiC_x$/Cu master alloy, which possess an average dimension of 100 nm, are uniform and spherical in shape.

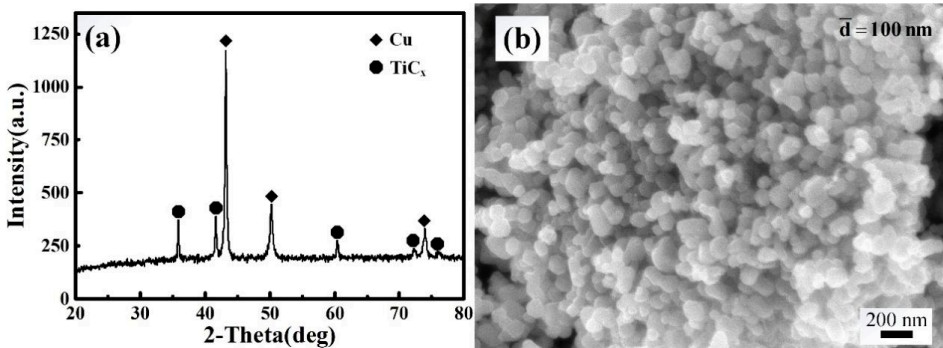

**Figure 2.** (**a**) XRD pattern of the $TiC_x$/Cu master alloy, (**b**) extracted $TiC_x$ particles from $TiC_x$/Cu master alloy.

The microstructures of Cu–Cr–Zr matrix alloy and $TiC_x$/Cu–Cr–Zr composite materials are illustrated in Figure 3. Dendrites are a dominant morphological feature of the Cu–Cr–Zr matrix alloy, manifested by coarse grains and uneven grain size, as shown in Figure 3a, whereas those $TiC_x$/Cu–Cr–Zr composites containing 2 wt.% and 4 wt.% $TiC_x$ have an equiaxed grain morphology with a uniform dimensional distribution, as shown in Figure 3b,c. In comparison with the matrix alloy, the composites have more uniform and denser structures. With increasing $TiC_x$ content, the grain morphologies of the composites transform from dendritic into equiaxial grain, with grain sizes refined considerably. Both the change in the grain morphology and the reduction in the grain size are related to the in situ addition of $TiC_x$ particles.

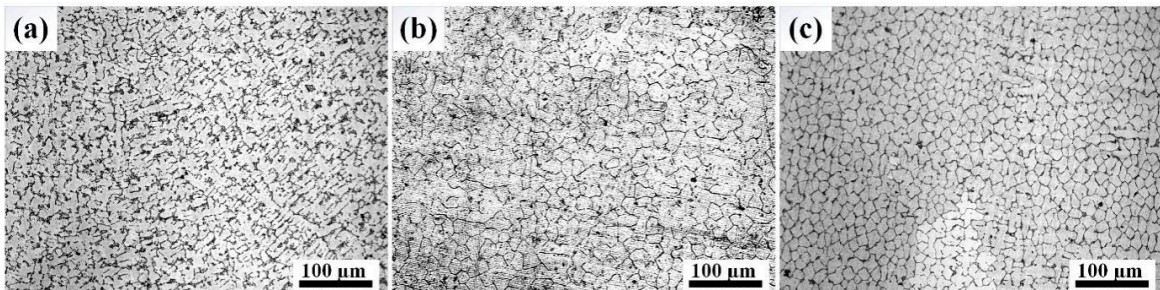

**Figure 3.** Grain morphology in the (**a**) Cu–Cr–Zr matrix alloy, (**b**) 2 wt.%, (**c**) 4 wt.% $TiC_x$/Cu–Cr–Zr composites.

Figure 4 is the microstructure of Cu–Cr–Zr alloy, 2 wt.% and 4 wt.% $TiC_x$/Cu–Cr–Zr composites. As shown in Figure 4a, grain shape is not obvious and the grain boundary range is relatively large. Whereas the $TiC_x$/Cu–Cr–Zr composites have an obvious microstructure and grain boundaries with a relatively uniform grain dimension. Furthermore,

the grain dimensions of the composites with a TiC$_x$ content of 4 wt.% are smaller than those of 2 wt.%. A large amount of chained particulate matter near the grain boundaries and a large distribution of particles are visible in the grains of the 4 wt.% TiC$_x$/Cu–Cr–Zr composites, as illustrated in Figure 4d.

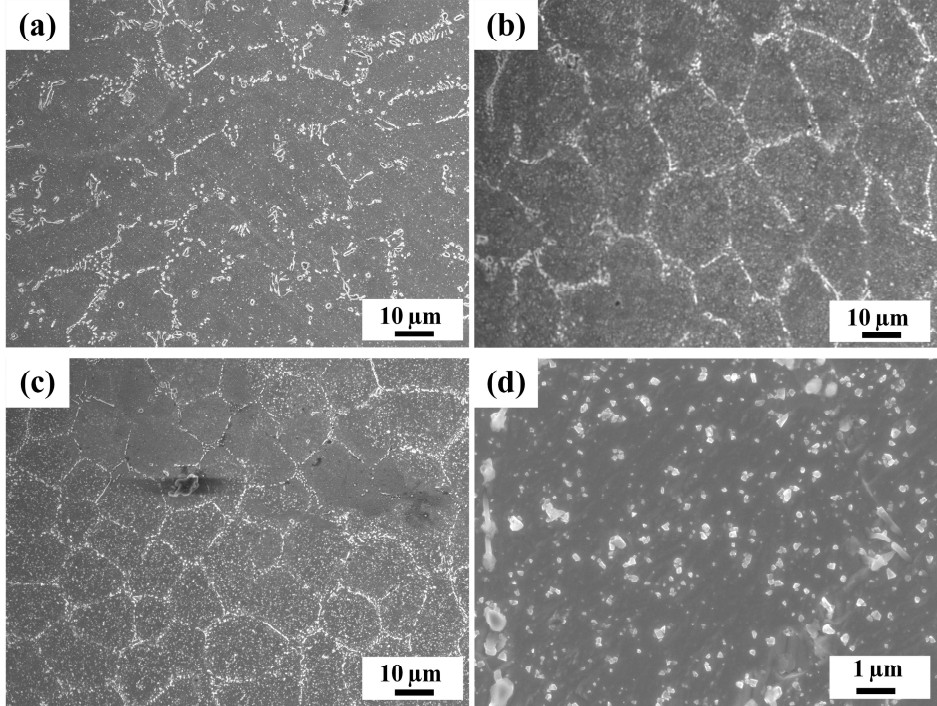

**Figure 4.** Microstructure of (**a**) Cu–Cr–Zr alloy, (**b**) 2 wt.% TiC$_x$/Cu–Cr–Zr composites, (**c**) 4 wt.% TiC$_x$/Cu–Cr–Zr composites, and (**d**) enlarged image of (**c**).

The binary phase diagram of Cu–Zr shows an approximate maximum solubility of Zr in Cu of 0.12 wt.% (927 °C), which is only about 0.01 wt.% at room temperature. During the solidification process, a large amount of Zr element and a considerable amount of Cu$_9$Zr$_2$ compound were precipitated. In addition, according to the binary phase diagram of Cu–Cr, the approximate maximum solubility of Cr in Cu is 0.7 wt.% (1076 °C), while it is only 0.03 wt.% at room temperature. Therefore, Cr elements were constantly precipitated with the decrease in the temperature in the solidification process. Some of the precipitated Cr elements were pushed to the grain boundary and gathered during the grain growth process, which showed the morphology of chain particles, as presented in Figure 4.

In order to explore the reason for the grain refinement phenomenon, lattice misfit ($\delta$) was utilized to determine the likelihood of heterogeneous nuclei, which was calculated via a mathematical model [27].

$$\delta_{(hkl)_n}^{(hkl)_s} = \frac{1}{3}\sum_{i=1}^{3} \frac{\left|d[uvw]_s^i \cos\theta - d[uvw]_n^i\right|}{d[uvw]_n^i} \times 100\% \tag{1}$$

where $(hkl)_s$ and $(hkl)_n$ are the low index crystal face; $[uvw]_s$ and $[uvw]_n$ are the low index crystal orientation in the $(hkl)_s$ and $(hkl)_n$; $d[uvw]_s$ and $d[uvw]_n$ are the atom spacings along the direction of the $[uvw]_s$ and $[uvw]_n$; $\theta$ is the angle between $[uvw]_s$ and $[uvw]_n$.

According to the results of the calculations, the lattice misfit of (111)$_{Cu}$ and (111)$_{TiC_x}$ is 19.8%, which means that TiC$_x$ is less likely to be a heterogeneous nucleus of the Cu alloy. However, the lattice misfit of (010)$_{Cu3Ti}$ and (111)$_{Cu}$ is 1.9%, and the lattice misfit of (100)$_{Cu3Ti}$ and (100)$_{TiC_x}$ is 1.2%. Therefore, it is speculated that the addition of TiC$_x$ changes the nucleation conditions of grains, and at the surface of the TiC$_x$ or TiC$_x$ clusters,

a layer of $Cu_3Ti$ forms that acts as heterogeneous nuclei for the grains. We speculated that heterogeneous nuclei are the main reason of changes in grain morphologies.

Figure 5 presents the microhardness and Brinell hardness of the Cu–Cr–Zr alloy and the $TiC_x$/Cu–Cr–Zr composites. The microhardness of matrix alloy composites with $TiC_x$ contents of 2 wt.% and 4 wt.% is 97.1 HV, 114.9 HV, and 119.7 HV, respectively, and the Brinell hardness is 75.1 HB, 84.1 HB, and 86.8 HB, respectively. The addition of $TiC_x$ nanoparticles effectively improves the hardness of the matrix alloy. Compared to the Cu–Cr–Zr alloy, the microhardness and Brinell hardness of 2 wt.% and 4 wt.% composites increase by 18.3% and 12.0%, 23.3% and 15.6%, respectively, with increasing $TiC_x$ content. It may be attributable to diffusely distributed $TiC_x$ particles within the composites, which can prevent the occurrence of plastic deformation. Based on load transfer strengthening, $TiC_x$ nanoparticles act as reinforcement, which indirectly bear the load applied on the materials. Moreover, the increasing number of $TiC_x$ particles lead to an increase in the amount of $TiC_x$ particles to bear the load; thus, the bearing capacity of the composite is enhanced. Meanwhile, fine grain strengthening also enhances the hardness of the composites. Consequently, an improvement in the hardness of composites is observed as the $TiC_x$ content increases.

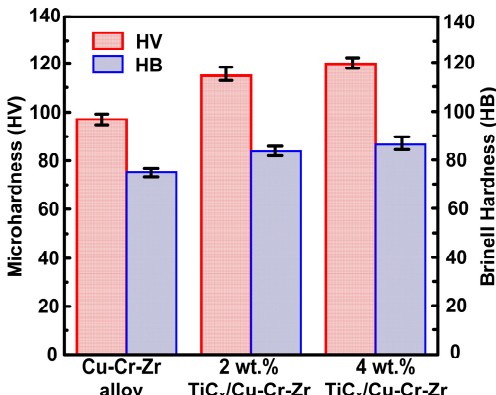

**Figure 5.** Microhardness and Brinell hardness of $TiC_x$/Cu–Cr–Zr composites with various $TiC_x$ contents.

Figure 6a–c shows the worn surface morphology of the Cu–Cr–Zr alloy and $TiC_x$/Cu–Cr–Zr composites with different $TiC_x$ contents with a 6.5 μm abrasive particle size and a 5 N load. An abrasive worn surface on the Cu–Cr–Zr alloy can be seen to be rough, with deep and wide surface furrows as shown in Figure 6a. With increasing $TiC_x$ content, abrasive worn surfaces of the composites become smooth, surface furrows become shallow and narrow, and the number of furrow ridges caused by plastic deformation decreases (Figure 6b,c). Figure 6d shows the abrasive wear rates for the Cu–Cr–Zr alloy and different $TiC_x$ content composites. The abrasive wear rates of Cu–Cr–Zr alloy, 2 wt.% and 4 wt.% $TiC_x$ content composites are 1.48, 1.35, and 1.31 ($10^{-10}$ $m^3$/m), respectively. In comparison to the Cu–Cr–Zr alloy, the composites showed a significant decrease in abrasive wear rate, which indicates that the incorporation of $TiC_x$ nanoparticles obviously improves the abrasive wear resistance of the Cu–Cr–Zr alloy. The addition of $TiC_x$ particles refines the microstructure, improving the hardness of the materials, which can reduce the depth of abrasive particle penetration into the materials and weaken the "plough and cut" phenomenon of the abrasive particles. Meanwhile, the nanoparticles act as reinforcement pinning in the matrix, which hinders the ploughing behavior of abrasive particles, preventing the deformation of materials. Therefore, the composites with 4 wt.% $TiC_x$ content exhibit the best abrasive wear resistance among the three materials.

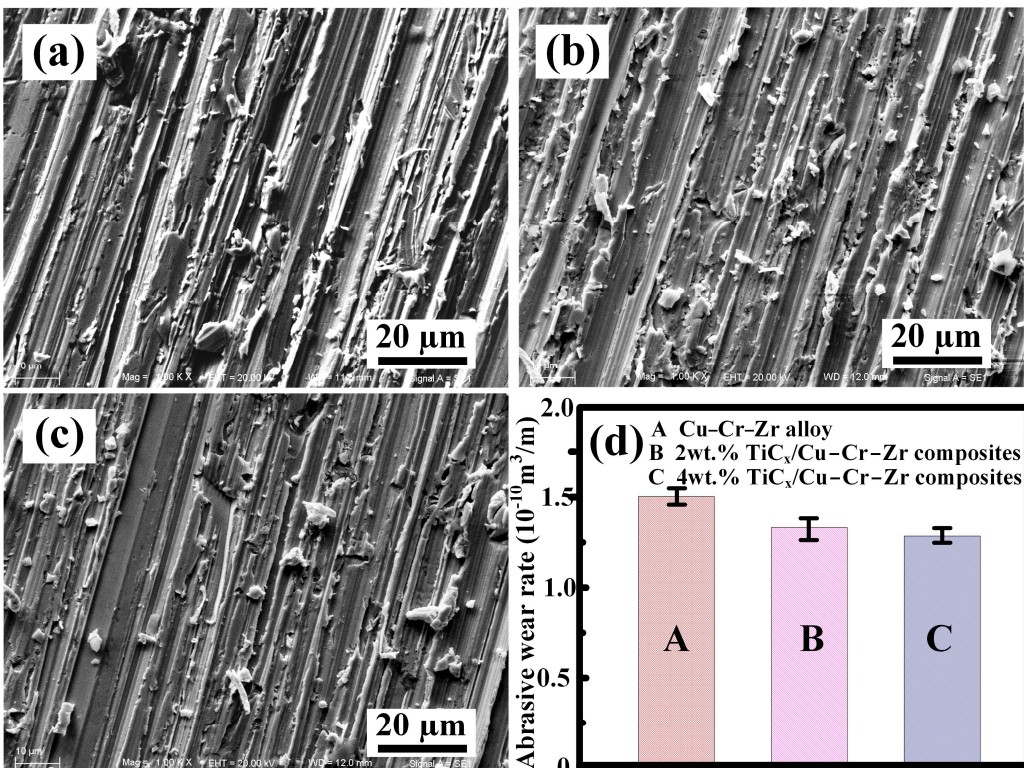

**Figure 6.** Worn surface of (**a**) Cu−Cr−Zr alloy, (**b**), 2 wt.% and (**c**) 4 wt.% TiC$_x$/Cu−Cr−Zr, (**d**) abrasive wear rate of corresponding materials.

It is believed that the increase in wear resistance and hardness for copper matrix composites is due to both the introduction of nano-TiC$_x$ particles and the Orowan mechanism. Moreover, nano-TiC$_x$ particles play a role as barriers during the dislocation movement process, which enhances the strength of the copper matrix composite. In contrast, the coefficient of thermal expansion mismatch and Taylor strengthening by modulus mismatch between the matrix and particle are also attributed to the strengthening of the composites' wear resistance [28].

The schematic diagram of the abrasive wear process for TiC$_x$/Cu−Cr−Zr composites with various TiC$_x$ contents is presented in Figure 7. During the abrasive wear process, as a consequence of abrasion and loads, the Cu−Cr−Zr alloy undergoes significant plastic deformation, which appears in the form of deep and wide furrows, as illustrated in Figure 7b. Shallower and narrower furrows on the worn surface of 2 wt.% TiC$_x$ composites are observed at identical abrasive and load conditions compared with the Cu−Cr−Zr alloy, as presented in Figure 7d. Composites with TiC$_x$ content of 4 wt.%, among the three materials, show the flattest worn surface under the same conditions, as shown in Figure 7f. TiC$_x$ nanoparticles, acting as the core of the heterogeneous nuclei, refine the grain size. Consequently, the grain morphology transformed from dendrite to equiaxial grain (Figure 7a,c,e). Grain refinement can improve the hardness of materials, which weakens the ploughing behavior of the abrasive particles by reducing their depth of penetration into the materials; therefore, the abrasive wear resistance of the composites is improved, as shown in Figure 7b,d,f. Moreover, the TiC$_x$ nanoparticles being added to the matrix alloy that are located at both grain boundaries and inside the grains corresponding to Figure 7 areas A and B, act as particle reinforcement, which fixes dislocations and prevents dislocation movement. The above reasons also contribute to the improvement of material. Thus, hardness and wear resistance were effectively improved by the addition of TiC$_x$ particles.

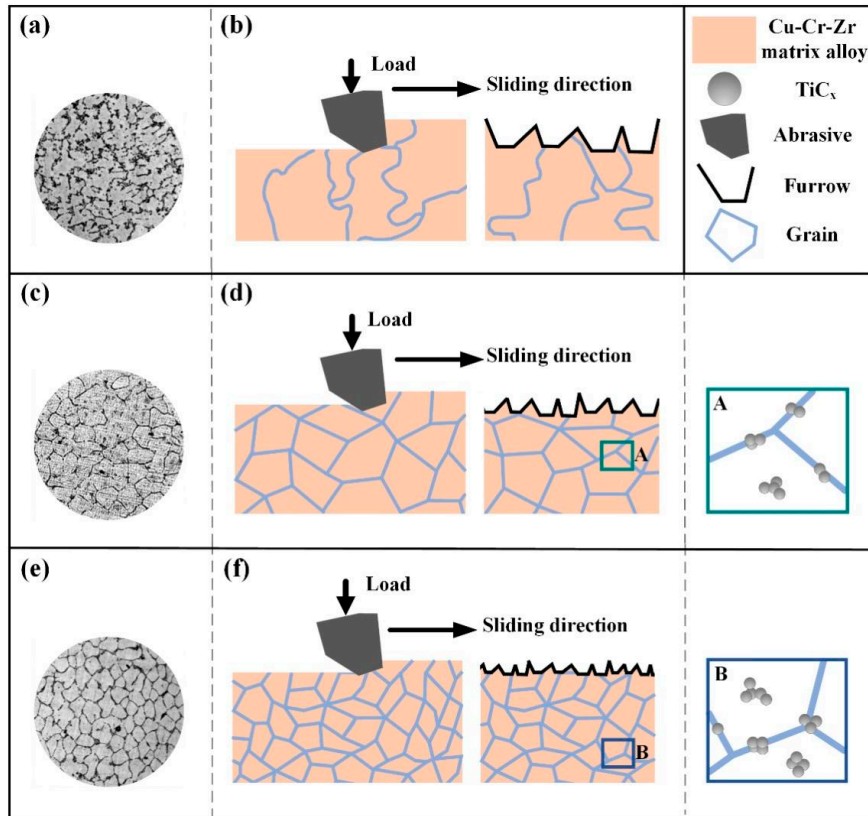

**Figure 7.** Schematic diagram of abrasive wear process for TiC$_x$/Cu–Cr–Zr composites with various TiC$_x$ contents. (**a**) metallographic morphology of Cu–Cr–Zr alloy (**b**) wear process of Cu–Cr–Zr alloy (**c**) metallographic morphology of 2 wt.% TiC$_x$/Cu–Cr–Zr composites (**d**) wear process of 2 wt.% TiC$_x$/Cu–Cr–Zr composites (**e**) metallographic morphology of 4 wt.% TiC$_x$/Cu–Cr–Zr composites (**f**) wear process of 4 wt.% TiC$_x$/Cu–Cr–Zr composites.

Figure 8 presents the electrical conductivity of Cu–Cr–Zr alloy, TiC$_x$/Cu–Cr–Zr composites with different TiC$_x$ contents. As indicated, the conductivity of the Cu–Cr–Zr alloy, 2 wt.% and 4 wt.% TiC$_x$ composites are 64.71%, 56.77%, and 52.93%, respectively (IACS). The incorporation of nanoparticles obviously refined the grain dimensions. The refined grain raises the quantity of grain boundaries and the scattering effect of the material on electrons is increased. In contrast, the incorporation of TiC$_x$ particles inhibits the migration of free electrons. The interface between the particle and matrix also enhances electron scattering, resulting in a reduction of 12.27% and 18.20% in electrical conductivity for composites with TiC$_x$ content of 2 wt.% and 4 wt.%, respectively, compared with the matrix alloy.

Thermal conductivity curves for TiC$_x$/Cu–Cr–Zr composites with various TiC$_x$ contents from 25 °C to 100 °C can be observed in Figure 9. It shows a gradual decrease in the thermal conductivity for the composite with increasing TiC$_x$ content at the same temperature. The coefficient of thermal conductivity of TiC$_x$ particles was significantly less than that of pure Cu. Therefore, the thermal conductivity of the material decreases as the rule Cu content decreases. Meanwhile, incorporating TiC$_x$ particles increased the bonding interface between the particles and matrix, enhancing the scattering effect by the interface upon heating in the composite, which is not conducive to heat conduction. In contrast, the thermal conductivity coefficient of composites drops as the temperature rises. The higher the temperature, the worse the thermal conductivity. Therefore, the thermal conductivity of the composite materials decreases gradually with the increase in temperature.

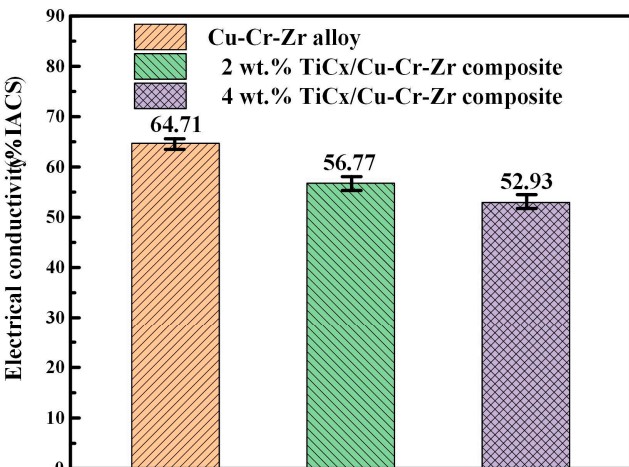

**Figure 8.** Electrical conductivity at various mass fractions of TiC$_x$/Cu–Cr–Zr composites.

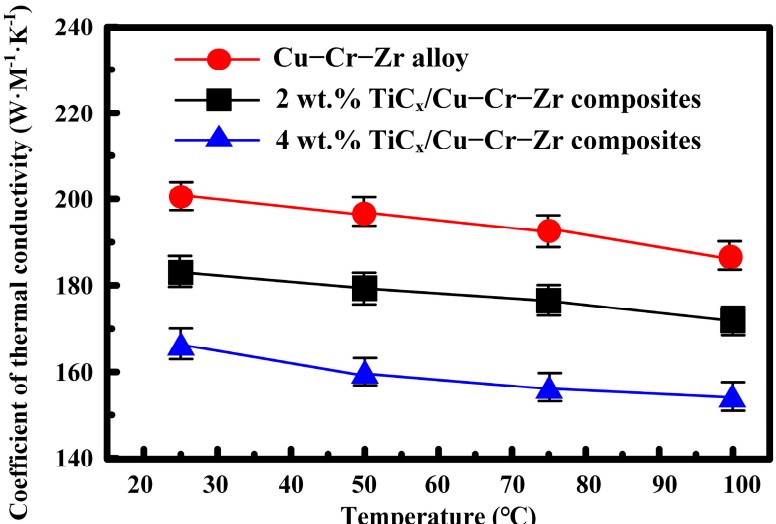

**Figure 9.** Coefficient of thermal conductivity for TiC$_x$/Cu–Cr–Zr composites with various TiC$_x$ contents at different temperatures.

Figure 10 presents the thermal expansion coefficient curves of the TiC$_x$/Cu–Cr–Zr composites with different TiC$_x$ contents from 25 °C to 400 °C. As shown, the thermal expansion coefficients of the materials gradual increase with increasing temperature for the three materials. The thermal expansion coefficients of the three materials are sensitive to TiC$_x$ content at the same temperature, with the sequence showing that the higher the TiC$_x$ content, the smaller the thermal expansion coefficient. It is indisputable that the thermal expansion coefficients of materials are determined by different external temperatures, the thermal expansion coefficient of the matrix metal, and the restraining effect of the reinforcing particles on the matrix. With the increase in temperature, the thermal expansion coefficients of all the three materials follow a similar trend; that is, as the temperature increases, the thermal expansion coefficients of the materials becomes larger. The higher the number of TiC$_x$ particles, the stronger the restraining effect on the matrix, causing the thermal expansion coefficient to decrease with the addition of TiC$_x$ particles.

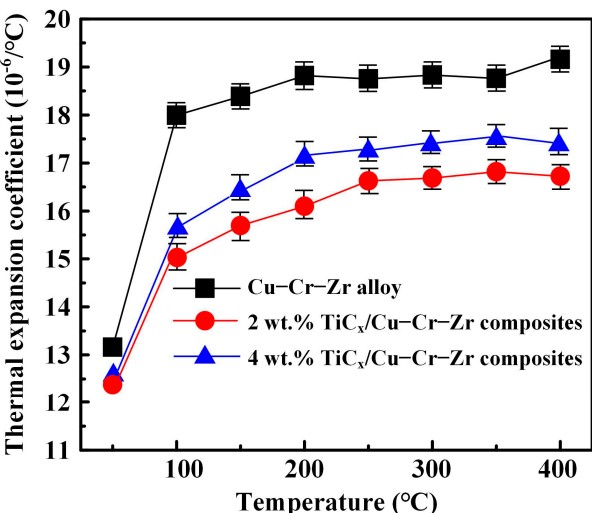

**Figure 10.** Thermal expansion coefficient at different temperatures for $TiC_x/Cu-Cr-Zr$ composites with various $TiC_x$ contents.

## 4. Conclusions

In this work, the microstructure evolution, abrasive wear, electrical conductivity, thermal conductivity, and thermal expansion coefficient of $TiC_x/Cu–Cr–Zr$ composites with reinforced in situ with different $TiC_x$ nanoparticle contents were investigated.

1. The in situ addition of $TiC_x$ nanoparticles transformed the microstructure of matrix alloy from dendrites to equiaxed crystal and refined the grain size.
2. Hardness and abrasive wear resistance were enhanced in the matrix alloy by adding $TiC_x$ particles. Microhardness and Brinell hardness of 4 wt.% $TiC_x/Cu–Cr–Zr$ composites are 23.3% and 15.6% higher, respectively, compared to the matrix alloy.
3. The 4 wt.% $TiC_x/Cu–Cr–Zr$ composite had the lowest abrasive wear rate ($1.31 \times 10^{-10}$ $m^3/m$) with the shallowest and narrowest surface furrows.
4. The thermal expansion coefficient, thermal conductivity, and electrical conductivity of the composites decreased with the increasing $TiC_x$ content. Electrical conductivity of 2 wt.% and 4 wt.% $TiC_x/Cu–Cr–Zr$ composites was reduced by 12.27% and 18.20%, respectively, compared with the matrix alloy.

**Author Contributions:** Conceptualization, D.Z. and Y.G.; methodology, P.L.; resources, Y.G.; data curation, D.Z. and Y.G.; writing—original draft preparation, D.Z.; writing—review and editing, D.Z. and X.H. All authors have read and agreed to the published version of the manuscript.

**Funding:** This work is supported by the Science and Technology Project of Jilin Provincial Education Department (JJKH20220098KJ).

**Institutional Review Board Statement:** Not applicable.

**Informed Consent Statement:** Not applicable.

**Data Availability Statement:** Not applicable.

**Acknowledgments:** This work is supported by Northeastern Electric Power University.

**Conflicts of Interest:** The authors declare no conflict of interest. The funders had no role in the design of the study; in the collection, analyses, or interpretation of data; in the writing of the manuscript; or in the decision to publish the results.

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
