# Peer review of "Abrasive Wear and Physical Properties of In-Situ Nano-TiCx Reinforced Cu–Cr–Zr Composites"

_coatings, doi:10.3390/coatings13071263_

Round 1
Reviewer 1 Report
The work still needs to be refined. I suggest a mandatory review of the following points to increase the quality of the paper.
1. Authors should show photos of the powders used.
2. Is the Cu-Cr-Zr alloy commercial? Provide the name of its manufacturer, as well as the manufacturers of powders.
3. The quality of Figs. 3 and 4 are poor.
4. In my opinion, the authors should take SEM pictures of the microstructures shown in Fig. 4. Eutectic is visible on the grain boundaries of the solid solution. Also, inside the grains, fine particles are visible. Higher magnification and better quality pictures are necessary.
5. The EDS maps are of poor quality and do not contribute anything. How long has the signal been collected? Fix this study.
6. The higher the content of TiC, the greater the number of crystallization nuclei. In addition, small particles are visible inside the grains, which I mentioned earlier. Please see the work at https://doi.org/10.1016/j.wear.2023.204834. There you will find information on strengthening copper sinter with nanoparticles.
7. There are no errors in Fig. 6.
8. How long was the friction wear test process carried out and at what temperature? There is no information.
9. There is no discussion or comparison to other works in the description of the results.
Recommendation: This manuscript in the presented form is not acceptable for publication in Coatings. A major revision is necessary.
Author Response
- Authors should show photos of the powders used.
Reply: Thanks for your suggestion. Ti and Cu powders have been used up, and we are now using different Ti and Cu powders, the process of purchasing from the manufacturer and conducting SEM testing takes more time. However, we performed TEM tests on carbon nanotubes, the TEM image is shown below:
Figure 1 TEM image of CNTs morphology
- Is the Cu-Cr-Zr alloy commercial? Provide the name of its manufacturer, as well as the manufacturers of powders.
Reply: Thanks for your suggestion. According to your suggestion, we have added the manufacturers of Cu-Cr-Zr alloy and the powders to manuscript. Some of the revised manuscript are listed as follow:
According to the vacuum gauge indication, heating was stopped after the combustion synthesis reaction, which resulted in the TiCx/Cu master alloy. Manufacturers of Ti, Cu, CNTs and Cu-Cr-Zr alloy are shown in Table 1. The chemical composition of the Cu-Cr-Zr alloy is illustrated in Table 2.
Table 1. Manufacturers of raw materials.
|
Raw material |
Manufacturers |
|
Ti |
Beijing Research Institute of Nonferrous Metals |
|
Cu |
Beijing Research Institute of Nonferrous Metals |
|
CNTs |
Chengdu Organic Chemistry Co., Ltd. Chinese Academy of Sciences |
|
Cu-Cr-Zr |
Shanghai Xinmeng Metal Materials Co., Ltd. |
- The quality of Figs. 3 and 4 are poor.
Reply: Thanks for your suggestion. Figs. in this manuscript were set 500*500 in pixel. We have checked that figs. were high quality before upload. The quality of Figs. become poor after upload. We have replaced Figs. 2, 3, 4 and 5 with higher quality pictures.
- In my opinion, the authors should take SEM pictures of the microstructures shown in Fig. 4. Eutectic is visible on the grain boundaries of the solid solution. Also, inside the grains, fine particles are visible. Higher magnification and better quality pictures are necessary.
Reply: Thanks for your suggestion. The images in Fig. 4 are SEM pictures of (a) Cu-Cr-Zr alloy, (b) 2 wt.% TiCx/Cu-Cr-Zr composites, (c) 4 wt.% TiCx/Cu-Cr-Zr composites. According to your suggestion, we have replaced a higher magnification and better quality picture in Fig.4(d). The revised manuscript is shown below:
Figure 4. Microstructure of (a) Cu-Cr-Zr alloy, (b) 2 wt.% TiCx/Cu-Cr-Zr composites, (c) 4 wt.% TiCx/Cu-Cr-Zr composites and (d) enlarged image of (c)
- The EDS maps are of poor quality and do not contribute anything. How long has the signal been collected? Fix this study.
Reply: Thanks for your suggestion. It is true that the EDS maps do not contribute anything. According to your suggestion, we have deleted the EDS maps and related description.
- The higher the content of TiC, the greater the number of crystallization nuclei. In addition, small particles are visible inside the grains, which I mentioned earlier. Please see the work at https://doi.org/10.1016/j.wear.2023.204834. There you will find information on strengthening copper sinter with nanoparticles.
Reply: Thanks for your suggestion. We have studied the work, we learned that the increase of hardness is related not only to the introduction of harder nano-TiCx particles into the copper matrix, but also is caused by the Orowan mechanism. In addition, nanoparticles constitute a barrier to dislocation movement, which influences the strengthening of the material. According to your suggestion, we have revised our manuscript, as follow:
The increase of hardness and wear resistance is related not only to the introduction of harder nano-TiCx particles into the copper matrix, but also is caused by the Orowan mechanism. In addition, nanoparticles constitute a barrier to dislocation movement, which influences the strengthening of the material. In addition, the increase in the wear resistance of metal composites is attributed to strengthening resulting from the Coefficient of Thermal Expansion mismatch between matrix and particles, and Taylor strengthening by modulus mismatch between matrix and particles [28].
- Adam P.; Piotr P.; Maciej T.; et al. Microstructure, mechanical properties and tribological behavior of Cu-nano TiO2-MWCNTs composite sintered materials. Wear. 2023; 522: 204834.
- There are no errors in Fig. 6.
Reply: Thanks for your suggestion. According to your suggestion, we have added errors in Fig. 6. We have revised our manuscript, as follow:
Figure 6. Microhardness and Brinell Hardness of TiCx/Cu-Cr-Zr composites with various TiCx contents.
- How long was the friction wear test process carried out and at what temperature? There is no information.
Reply: Thanks for your suggestion. According to your suggestion, we have added more detailed experimental procedures and parameters for the friction wear test process. The revised manuscript is as follow:
Both of Microhardness and Brinell hardness were measured 5 times respectively and averaged. Tests of abrasive wear were performed using a pin-disc machine with Al2O3 abrasive paper under loads 5 N with a wear distance of 24.78 m at room temperature. The samples were 4 mm in length, 4 mm in width, and 12 mm in height. The worn surfaces of the composites were observed by a scanning electron microscopy (SEM, Evo18, Carl Zeiss, Germany).
- There is no discussion or comparison to other works in the description of the results.
Reply: Thank you very much for your valuable advice, in other literatures, the properties of copper alloys are tested after heat treatment and deformation. This study is only based on the as-cast Cu-Cr-Zr alloy, and no follow-up experiments have been conducted. When the follow-up tests are completed, we will sort out relevant literatures for discussion and comparison.

Reviewer 2 Report
1) Kindly please enhance language standard in some places
2) One more keywords can be provided
3) How TiCx can modify the parent composites's nature? Provide more recent references which related to effects of carbide / effect of reinforcement in Zr contained alloy in the composites? Effect of Age-Hardening Temperature on Mechanical and Wear Behavior of Furnace-Cooled Al7075-Tungsten Carbide Composite; Effect of Scandium in Al-Sc and Al-Sc-Zr Alloys under precipitation strengthening mechanism at 3500C Aging; Optimization of Abrasive Water Jet Machining of SiC Reinforced Aluminum Alloy Based Metal Matrix Composites Using Taguchi–DEAR Technique; Effects of Silicon Carbide and Tungsten Carbide in Aluminium Metal Matrix Composites; Effect of B4C and MOS2 reinforcement on micro structure and wear properties of aluminum hybrid composite for automotive applications
4) What is nature of Cr and Zr in Cu alloy?
5) Figure 2-5 quality can be improved. The scale is visible in blurred version.
6) Figure 6 and 11can be given with error bar indication to show the standard error, if possible
7) The main core theme results can be provided point by point in conclusion section.
Can be improved in some places.
Author Response
- Kindly please enhance language standard in some places
Reply: Thank you very much for your valuable advice, so that the paper can be improved. We have modified the English expression in the Manuscript.
- One more keywords can be provided
Reply: Thanks for your suggestion. According to your suggestion, we have revised our manuscript, as follow:
Keywords: TiCx/Cu; in situ nano-TiCx; abrasive wear resistance; physical properties
3) How TiCx can modify the parent composites' nature? Provide more recent references which related to effects of carbide / effect of reinforcement in Zr contained alloy in the composites? Effect of Age-Hardening Temperature on Mechanical and Wear Behavior of Furnace-Cooled Al7075-Tungsten Carbide Composite; Effect of Scandium in Al-Sc and Al-Sc-Zr Alloys under precipitation strengthening mechanism at 3500C Aging; Optimization of Abrasive Water Jet Machining of SiC Reinforced Aluminum Alloy Based Metal Matrix Composites Using Taguchi–DEAR Technique; Effects of Silicon Carbide and Tungsten Carbide in Aluminium Metal Matrix Composites; Effect of B4C and MOS2 reinforcement on micro structure and wear properties of aluminum hybrid composite for automotive applications.
Reply: Thanks for your suggestion. According to your suggestion, we have provided several recent references in our revised manuscript, as follow:
- Liu S.; Wang Y.; Muthuramalingam T.; et al. Effect of B4C and MOS2 reinforcement on micro structure and wear properties of aluminum hybrid composite for automotive applications. Compos Part B. 2019; 176: 107329.
- Xue N.; Liu W.; Zhu l.; et al. Effect of Scandium in Al–Sc and Al–Sc–Zr Alloys Under Precipitation Strengthening Mechanism at 350 °C Aging. Met Mater Int. 2020; 27: 5145-5153.
- Adam P.; Piotr P.; Maciej T.; et al. Microstructure, mechanical properties and tribological behavior of Cu-nano TiO2-MWCNTs composite sintered materials. Wear. 2023; 522: 204834.
4) What is nature of Cr and Zr in Cu alloy?
Reply: Thanks for your suggestion. In our opinion, in the alloy, the Cr element first forms a lattice solid solution, and the Zr element forms a Cr-Zr eutectoid at the grain boundary, thus forming a more stable structure. When the alloy is heat treated, the Cr element in the alloy will gradually diffuse into the matrix, and eventually form Cu-Cr subeutectoid phase. At the same time, Zr element is repulsed to the interface of the subeutectoid phase to form Cr-Zr eutectoid. This process will lead to a decrease in the concentration of Cr in the alloy, and the formation of Cr-Zr eutectoids will lead to an increase in the strength of the alloy.When the alloy is aged, the Cr and Zr elements in the alloy will become solid solutions, and the nanocrystalline grains will be formed in the collective. These nanocrystals and the Cr and Zr elements in the alloy work together to improve the strength of the alloy.
5) Figure 2-5 quality can be improved. The scale is visible in blurred version.
Reply: Thanks for your suggestion. Figs. in this manuscript were set 500*500 in pixel. We have checked that figs. were high quality before upload. The quality of Figs. become poor after up load. We have replaced Figures 2-5 with higher quality pictures.
6) Figure 6 and 11can be given with error bar indication to show the standard error, if possible
Reply: Thanks for your suggestion. According to your suggestion, we have added errors in Fig. 6. We have revised our manuscript, as follow:
Figure 5. Microhardness and Brinell Hardness of TiCx/Cu-Cr-Zr composites with various TiCx contents.
Figure 8. Electrical conductivity at various mass fractions of TiCx/Cu-Cr-Zr composites.
7) The main core theme results can be provided point by point in conclusion section.
Reply: Thanks for your suggestion. According to your suggestion, we have revised our manuscript, as follow:
Conclusions
In this work, the microstructure evolution, abrasive wear, electrical conductivity, thermal conductivity and thermal expansion coefficient of TiCx/Cu-Cr-Zr composites with different in-situ TiCx nanoparticle contents were investigated.
- The addition of in-situ TiCx nanoparticles made the microstructure of matrix alloy transformed from dendrites to equiaxed crystal and refined the grain size.
- Hardness and abrasive wear resistance were enhanced of matrix alloy by adding TiCx particles. Microhardness and Brinell hardness of 4 wt.% TiCx/Cu-Cr-Zr composites are 23.3% and 15.6% higher, respectively, compared to the matrix alloy.
- 4 wt.% TiCx/Cu-Cr-Zr composite showed the lowest abrasive wear rate (1.31×10-10 m3/m) with the shallowest and narrowest surface furrow.
- The thermal expansion coefficient, thermal conductivity and electrical conductivity of the composites decreased as the increasing TiCx content. Electrical conductivity of 2 wt.% and 4 wt.% TiCx/Cu-Cr-Zr composites was reduced by 12.27% and 18.20%, respectively, compared with matrix alloy.

Round 2
Reviewer 1 Report
The authors have taken into account the comments of the reviewer and made corrections in this article. They have responded to all the comments from the reviewer.
There is an error in the literature. In item 28 there are incorrectly spelled names - there is: Adam P.; Piotr P.; Maciej T.; et al. - it should be Piasecki A.; Paczos P.; Tulinski M., et al.
After this minor correction, I recommend this manuscript for publication in Coatings.
Author Response
Response to Reviewer 1 Comments
- The authors have taken into account the comments of the reviewer and made corrections in this article. They have responded to all the comments from the reviewer.
There is an error in the literature. In item 28 there are incorrectly spelled names - there is: Adam P.; Piotr P.; Maciej T.; et al. - it should be Piasecki A.; Paczos P.; Tulinski M., et al.
Reply: Thanks for your suggestion. According to your suggestion, we have revised our manuscript, as follow:
- Piasecki A.; Paczos P.; Tulinski M.; et al. Microstructure, mechanical properties and tribological behavior of Cu-nano TiO2-MWCNTs composite sintered materials. Wear. 2023; 522: 204834.

Reviewer 2 Report
The comments have been addressed
Minor revision is needed
Author Response
The comments have been addressed
Reply: Thank you very much for your valuable advice, so that the paper can be improved.